



# Apparent dust size discrepancy in aerosol reanalysis in north African dust after long-range transport

Samantha J. Kramer[1], Claudia Alvarez[1], Anne Barkley[1], Peter R. Colarco[2], Lillian Custals[1], Rodrigo Delgadillo[1], Cassandra Gaston[1], Ravi Govindaraju[2], and Paquita Zuidema[1]

[1]University of Miami, Rosenstiel School of Marine and Atmospheric Science, Miami, FL, USA
[2]NASA Goddard Space Flight Center, Greenbelt, MD, USA

**Correspondence:** Paquita Zuidema (pzuidema@miami.edu)

**Abstract.** North African dust reaches the southeast United States every summer. Measurements taken in Miami, Florida indicate that more than one-half of the surface dust mass concentrations reside in particles with diameters less than 2.1 $\mu$m, while vertical profiles of micropulse lidar depolarization ratios show dust reaching above four km during pronounced events. These observations are compared to the representation of dust in the MERRA-2 aerosol reanalysis and closely-related GEOS-

5 5 Forward Processing (FP) aerosol product, both of which assimilate satellite-derived aerosol optical depths using a similar protocol and inputs. These capture the day-to-day variability in aerosol optical depth well, in a comparison to an independent sun-photometer-derived aerosol optical depth dataset. Measured near-surface dust mass concentrations slightly exceed model values, with most of the modeled dust mass in diameters between 2-6 $\mu$m. Modeled-specified mass extinction efficiencies equate light extinction with approximately three times as much aerosol mass, in this size range, compared to the measured

dust sizes. GEOS-5 FP surface-layer sea salt mass concentrations greatly exceed observed values, despite realistic winds and relative humidities. In combination, these observations help explain, why, despite realistic total aerosol optical depths, 1) free-tropospheric model volume extinction coefficients are lower than those retrieved from the micro-pulse lidar, suggesting too low model dust loadings, and 2) model dust mass concentrations near the surface are higher than those measured. The modeled vertical distribution of dust, when captured, is reasonable. Large, aspherical particles exceeding the modeled dust sizes are

also occasionally present, but dust particles with diameters exceeding ten $\mu$m contribute little to the measured total dust mass concentrations after such long-range transport. A further integrated assessment is needed to confirm this study's interpretations.

## 1 Introduction

Africa contributes almost 50% of the world's dust emissions (Huneeus et al., 2011), with easterly winds transporting the dust

over the Atlantic subtropical belt throughout the full seasonal cycle. The impacts of dust span from climate-relevant time scales, which primarily focus on dust's radiative effects, to the shorter time scales important for human endeavours. The latter include dust storms capable of closing airports, disrupting military exercises and agriculture, and impacting human health. The amount





of dust mass in diameters less than 2.5 $\mu$m in particular is important for air quality, as these can penetrate human lungs (Wilson and Spengler, 1996), setting the diameter threshold used by the United States Environmental Protection Agency in meeting the conditions of the Clean Air Act. Dust mass concentrations in this size range frequently exceed established standards in the Eastern Caribbean (Garrison et al., 2014), if rarely in the southeastern United States (Prospero, 1999b).

The desire for short-term information on dust emissions and their transport has encouraged the dissemination of forecasts through, for example, the World Meteorological Organization's Sand and Dust Storm Warning Advisory and Assessment System. More accurate shorter-term predictions have resulted from a further approach, in which the aerosol forecasts are adjusted by assimilating observed aerosol optical depths ($\tau_{aer}$), primarily derived from satellite. The data assimilation allows global aerosol models to improve the spatial distribution of aerosol plumes and is particularly valuable further away from

aerosol source regions, where models tend to underestimate $\tau_{aer}$ (Kim et al., 2014; Evan et al., 2014).

A leading global aerosol model also used for operational forecasts is the Goddard Earth Observing System Model Version 5 (GEOS-5), which uses the Goddard Chemistry Aerosol Radiation and Transport (GOCART;  Ginoux et al., 2001) scheme. The operational aerosol forecast GEOS-5 Forward Processing (FP) model relies on a slightly more mature version of the GEOS-5 model used to produce the arguably better-documented Modern-Era Retrospective analysis for Research and Application-

Version 2 (MERRA-2;  Randles et al., 2017). The assimilated $\tau_{aer}$, with which the simulated aerosol loadings are adjusted, primarily come from the Moderate Resolution Imaging Spectroradiometer (MODIS) satellite sensor. The $\tau_{aer}$ biases produced by clouds and inhomogeneous observations is corrected using a neural network based on reference surface-based Aerosol Robotic Network (AERONET; Holben et al., 1998) $\tau_{aer}$s. The MERRA-2 Reanalysis shows no global bias relative to the surface-derived AERONET $\tau_{aer}$ at 550 nm (Gelaro et al., 2017). The assimilation reduces the presence of spurious trends

in MERRA-2 (Gelaro et al., 2017) and clearly improves depicted atmospheric aerosol loadings compared to those in non-aerosol-assimilating simulations (Buchard et al., 2017). Importantly for dust transport, the assimilation corrects for a tendency of GEOS-5 to remove dust too quickly from the atmosphere (Buchard et al., 2017), presumably through either excessive sedimentation of the larger particles, or excessive scavenging by clouds. The GEOS-5 FP/MERRA-2 products are similar enough that they are used interchangeably within this study, reflecting their incorporation at different times of the study.

The assimilated $\tau_{aer}$ must be redistributed into a vertical structure of the aerosol mass mixing ratio, differentiated between diverse aerosol sources, of which two (dust and sea salt) are further distributed into five different size ranges. Mass extinction efficiencies are developed using radiative transfer theory for each of the aerosols and parameterized. These allow the second moment of the size distribution, namely the vertically-resolved volume extinction coefficient, to be interpreted as the third moment of the size distribution, or the aerosol mass mixing ratio. The assimilation corrects the model-specified aerosol mass

mixing ratios to approximately match the observed $\tau_{aer}$, so that the spatial distribution of the aerosol is improved, with the aerosol source and vertical distribution primarily set by the aerosol parameterizations and model physics. The model physics driving the convective transport, cloud microphysics and mixing influence how much of the aerosol is lofted further, how much is captured by precipitation, how much mixes into the boundary layer, and how much is deposited back to the surface. The particle size and vertical structure distributions are not necessarily independent, as smaller particles will be lofted higher and

larger particles will settle out of the atmosphere more quickly. The aerosol size parameterizations therefore have ramifications





for the subsequent transport, the resulting direct radiative effect, and the model aerosol deposition rates. In one comparison of dust vertical profiles, an Aerosol Comparison between Observations and Models (AeroCom) intercomparison of five emission-based models found significant diversity in the modeled vertical dust profiles, with large discrepancies compared to space-based lidar observations (Koffi et al., 2016).

Further understanding of global aerosol model forecasts can be gained from their assessment using *in-situ* and space-based observations. This study assesses the depiction of dust properties after long-range transport within the GEOS-5 FP forecasts and MERRA-2 reanalysis at a location slightly offshore of Miami, Florida, or over 20,000 km away from the source region. Dust is the dominant aerosol present in the Miami, Florida atmosphere during the summer months (Fig. 1), with sea salt a distant second, even at a coastal site with a consistent sea-salt presence throughout the year (Prospero, 1999a, replotted as the

inset in Fig. 1)). Anthropogenic (nitrate, non-sea-salt sulfate) and biogenic (potassium, calcium) mass concentrations are low (Fig. 1). As detailed in many previous studies and easily verified with satellite imagery and GEOS-5 forecasts, most of the dust originates from northern Africa, with the July maximum in the dust mass concentrations at Miami reflecting a combination of the easterly winds over the remote tropical Atlantic and more southerly winds over the Caribbean (Kramer et al., 2019), with less aerosol removal by precipitation in July than in August. Variability in sea salt mass concentrations can be more

directly related to local surface wind speeds. Overall, the relatively simple summer aerosol environment of Miami, consisting of primarily dust and sea salt, simplifies this study.

## 2   Background and Datasets

Primary observations include ground-based micropulse lidar (MPL) depictions of the dust vertical structure, surface measurements of bulk and size-resolved dust mass concentrations and sea salt mass concentrations. The daily resolved measurements

span three consecutive Miami dust seasons, namely June through September of 2014, 2015 and 2016. The filter and lidar data are collected at the Rosenstiel School of Marine and Atmospheric Science campus of the University of Miami, located on Virginia Key at 25.73°N, 80.16°W, approximately four km east of the Florida mainland. Filter measurements of the bulk mass concentration are made daily from May-September, with more details on the protocol available within Zuidema et al. (2019); these basically follow Prospero (1999a). Further analysis, in support of this study, included analysis of extracts containing the

major soluble inorganic ions using the samples from July and August of 2014, primarily to ascertain the sodium mass concentrations. This used flame atomic absorption to determine sodium ($Na^+$) and suppressed ion chromatography to determine chloride ($Cl^-$) (Savoie et al., 1989) along with nitrate ($NO_3^-$) and sulfate ($SO_4^-$). The sea salt concentration is computed from the sodium concentration multiplied by 3.256, following Prospero (1999a). Filter-based dust mass measurements discriminated by size are also collected for select days in 2016 encompassing a strong synoptic event. Scanning electron microscopy imaging

is additionally performed on select filters from this time period. Throughout, the daily dust bulk mass measurements at the surface are matched to the higher time-resolution of lidar measurements. The dust layers and their vertical extent are identified using the polarization capability of aMPL, and extinction profiles are retrieved from the lidar backscattered intensities for select cases following Delgadillo et al. (2018), with details relevant to this study provided below.



## 2.1 Size-resolved dust mass concentrations

Dust size distributions were measured on 17 days in 2016 using a Series 230 High Volume Cascade Impactor: Multi-Stage Particulate Size Fractionator manufactured by Tisch Environmental, Inc, following Li-Jones and Prospero (1998). The instrument was located directly below the bulk filters, using a secondary calibrated airflow. Air flow through three aluminum slots

discriminated for particle aerodynamic diameters of 2.1, 4.2, and 10.2 microns (see also Table 1) with one final plate capturing the remaining smaller sizes. The asphericity of dust may introduce an error in the interpretation of each size range, since the impactor selects for the particle size based on only the two horizontal dimensions. The size-resolved dust masses are thereafter extracted identically to those on the bulk filters.

The total dust masses summed from the sized impactor samples agree well with the dust masses from the bulk samples

(Fig. 2). Li-Jones and Prospero (1998) report dust masses summed from cascade impactors that are 25.4% to 53.5% less than those from bulk samples. A bias towards smaller summed dust masses from impactor samplers is also documented in a more comprehensive assessment of sizing techniques (Reid et al., 2003a). This suggests the good agreement documented here may be fortuitous, for example, through a difference between the true and assumed air flow. Nevertheless, the robust correlation between the two independent measures provides confidence in the size-resolved dust mass concentrations.

## 2.2 Scanning electron microscopy imaging

Scanning electron microscopy imaging was done on filters from August 1 and 2 of 2016, to confirm the size of the largest dust particles and provide some insight into dust geometry and composition. The imaging was conducted on a 1 cm by 1 cm area of the original filter. The samples were coated with palladium in a Cressington-108 Sputter Coater and then imaged with a Phillips XL-30/ESEM-FEG at 20kV at the University of Miami's Center for Advanced Microscopy.

## 2.3 Micropulse lidar

The MPL depolarization capability relies on a toggling between two modes using a polarizing beam splitter, with the ratio of the perpendicularly-polarized to the parallel-polarized radiation at 532 nm wavelength forming the volume linear depolarization ratio ($\delta_v$). $\delta_v$ can be used to distinguish round particles, such as liquid drops and marine aerosols, from particles with distinctive aspherical shapes such as dust. In the free troposphere when dust is the dominant aerosol, the measured $\delta_v$ is almost entirely

due to dust. For pure near-source dust, Freudenthaler et al. (2009) report a particle linear depolarization ratio ($\delta_p$; this includes a correction for the measured molecular contribution) ratio of $0.31 \pm 0.03$ at 532 nm. $\delta_p$ values for north African dust transported to the Caribbean can be slightly lower (0.30 within Burton et al. (2015) and $0.28 \pm 0.02$ within Haarig et al. (2017)). The most likely cause is a shift to smaller sizes with a lower depolarization ratio, although some reduction in aerosol asphericity may also occur from chemical aging resulting in the accumulation of soluble material.

$\delta_v$ is a volume measurement, and when dust is mixed with other aerosols such as sea salt or sulfate, such as will occur in the boundary layer, the measured total $\delta_v$ values will be reduced. This can be understood through $\delta_v \sim \frac{d_{dust}X}{1-dX}$ where $d_{dust} = \frac{\delta_v}{1+\delta_v}$ and $X$ is the dust's contribution to the backscattered radiation when no other depolarizing aerosols present (Sugimoto and Lee,





2006). The $\delta_p$ of marine salt is approximately 0.05 (Burton et al., 2012), or much lower than that of dust. $\delta_v$ values for north African dust after long-range transport range from 0.20 to 0.30 (Kanitz et al., 2014), attributed to mixing with other aerosols. Cloud particles, both liquid and ice, can be distinguished from dust by their higher volume backscatter intensities, with visual imagery from collocated cameras used to discriminate cloudy from cloud-free conditions (Delgadillo et al., 2018).

The Miami MPL was calibrated by the vendor, Sigma Space, prior to its delivery to Miami in 2013, and its calibration and overlap function were checked in 2016 using a mirror to send and receive the lidar laser beam through the atmosphere horizontally, on a day chosen for its clear, horizontally-homogeneous conditions. The correction for the near-field overlap occurring within the laser beam is most significant below one km (Delgadillo et al., 2018), and has less influence on measurements of free tropospheric aerosol. The depolarization ratio is not sensitive to the overlap correction. The lidar is kept in a temperature-
controlled environment, which reduces fluctuations in the lidar backscatter-to-extinction ratio, and is located approximately 100 m away from where the filter measurements are made. Increases in the volume depolarization ratio from 2015 to 2016 coincide with the incorporation of a new laser diode pump in May, 2016. This will increase the laser beam intensity, increasing the beam's ability to penetrate into the atmosphere, increasing the signal to noise of the signal in both channels (cross- and co-polarization), and ultimately increasing the $\delta_v$ value. As such, the volume depolarization ratios reported here serve more as
a qualitative indicator for the presence of the dust, than a quantitative measure, although more intense dust events are typically associated with higher volume depolarization ratios (at the same laser intensity).

    The MPL measures a backscattered intensity, which can only be related to the more physically-relevant aerosol volume extinction coefficient through a retrieval that is under-constrained. A sun-photometer-derived aerosol optical depth provides a vertically-integrated constraint on the extinction retrieval (Delgadillo et al., 2018). The two sun photometers, located near the
lidar on a rooftop of a three-story building, are part of the Aerosol Robotic Network (AERONET; Holben et al., 1998). The Miami sun photometer version 2 data were calibrated and cloud-filtered (level 2) for 2015, and solely cloud-filtered during 2014 and 2016 (level 1.5). The retrieval also produces a column-average ratio of the backscattered intensity to extinction, known as the lidar ratio. This will also vary with particle size. Because the vertical column above the lidar also contains sea salt in the lower atmosphere, the column-average lidar ratio is not that of the dust in the free troposphere. Lidar ratios range
from 40-60 sr for dust and 15-25 sr for sea salt (Burton et al., 2012); a column-average lidar ratio less than that appropriate for dust is consistent with an overestimate in the lidar-retrieved extinction. The optical properties of dust are not thought to vary with relative humidity (Maring et al., 2003a), but those for sea salt, a hydrophillic aerosol, will. We estimate the overall uncertainty in the retrieved lidar extinction conservatively with a factor of three. The lidar vertical resolution is 30 m and the time resolution is 15 seconds (done to facilitate detection of the small clouds common to Miami).

**2.4   GEOS-5 FP/MERRA-2**

The GOES-5 model possesses 72 vertical layers, of which the mid-level of the lowest layer is at approximately 69 meters. Data from the lowest model layer are compared to the measured surface aerosol concentrations. The aerosol assimilation occurs eight times per day, and the output frequency of the full three-dimensional aerosol field is also every three hours. We primarily consider the dust and sea salt contributions. The GOCART dust emission parameterization depends on a source function, the





near-surface wind speeds, and soil moisture (Ginoux et al., 2001). The dust sizes, prescribed at emission, can evolve thereafter during transport, with sedimentation the primary process capable of altering the dust size distribution. The parameterization of the dust mass distribution by size approximately follows that of Tegen and Lacis (1996) and contains five size categories, extending to a maximum diameter of 20 $\mu$m. These are indicated in Table 1 along with the density and mass extinction efficiency

corresponding to each size category. Spheroidal particles are assumed, with more details on the dust parameterization available in Colarco et al. (2010). The dust is assumed to not be hygroscopic.

The flux of sea salt off of the ocean surface is also parameterized into five size bins, but the size is allowed to vary with the surface wind speeds and sea surface temperature, and the sea salt can undergo hygroscopic growth as a function of the relative humidity. The sea salt parameterization is adapted from Gong (2003) and described further in Chin et al. (2002) and Bian et al.

(2019).

Mass extinction efficiencies - the ratio of extinction to mass, specific for each diameter size range (see Table 1 for the dust mass extinction efficiencies) - connect the model aerosol mass mixing ratios to the assimilated aerosol optical depth. The clear-sky $\tau_{aer}$ values at a wavelength of 550 nm are primarily assimilated from the Moderate Resolution Imaging Spectrora- diometer (MODIS) and secondarily the Multi-Angle Imaging SpectroRadiometer (MISR) satellite instruments (Randles et al.,

2017). Data from the surface-based AERONET sites are no longer assimilated (Randles et al., 2017), allowing the Miami sun photometer data to independently validate the GEOS-5 FP $\tau_{aer}$s. The lowest model level is assumed to represent surface values.

## 2.5 HYSPLIT back trajectories

Daily average back-trajectories explore the differences in the integrated air flow for days with high and low dust mass concen-

20 trations. The calculations rely on the NOAA Air Resources Laboratory HYSPLIT model (Stein et al., 2015). These extend for 300 hours (12.5 days) and are driven by the NCEP Reanalysis Climate Diagnostics Center-1 product. Each day's trajectory is initialized at 00Z and at an altitude of 2000 m. This is a robust height for the presence of dust above Miami, as indicated later. There is some evidence that the NCEP Reanalysis lower tropospheric winds may be weaker than observed (e.g., Adebiyi et al., 2015), which would increase the transport time, but this is not investigated further for this study.

## 25  3  Overview

A time series of the primary measurements displays the daily surface-based filter-derived dust mass concentrations and the corresponding vertical structure as captured by the lidar volume depolarization ratio, for each of the three dust seasons (Fig. 3). The lidar measurements indicate that dust is often prevalent in the boundary layer, typically extends up to 2-3 km, and occa- sionally rises above 4 km but not above 5 km. The depolarization ratios indicate that when dust is present, it is typically also

present within the boundary layer. This is corroborated by the filter measurements; only during mid-to-late August 2015 is no dust detected within the atmospheric column by either the filters or the lidar. The synoptic and year-to-year variability is clear, with the highest dust mass concentrations occurring on just a few select days of each summer.





The back-trajectories indicate the crucial role of the north Atlantic subtropical high in guiding the atmospheric flow towards Miami (Fig. 4(a-c)), explored further in Kramer et al. (2019). Many back-trajectories intersect the coastal region off of North Africa, with few continuing further eastward over the African continent. This may indicate a strong influence from the northerly coastal flow on the eastward side of the north Atlantic subtropical high. The HYSPLIT back-trajectories disregard explicit

mixing with environmental air, but a back-trajectory with a large meridional component would imply more mixing of the dusty air with air emanating from the northeast Atlantic and Europe. That this occurs is consistent with the co-existence of anthropogenic aerosols with dust (Fig. 1). The separate influences of African versus Atlantic air on Miami and Caribbean thermodynamic profiles is also noted in Dunion (2011).

    After twelve days, the integration confidence in the trajectories leading to the original location of air parcel is debatable,

therefore we focus more on the trajectory characteristics over the Atlantic. Back-trajectories associated with days with dust mass concentrations exceeding 20 $\mu$g m$^{-3}$ are typically more zonal, suggesting less mixing with air originating from the northeastern Atlantic. Back trajectories associated with days with high dust mass concentrations also have a slight preference for a more northerly location. This may indicate less aerosol removal by precipitation en route (the northern boundary of the JJA intertropical convergence zone is at approximately 10° N). A notable feature of the back-trajectories on days with low

($< 1$ $\mu$g m$^{-3}$) and moderate (1 - 20 $\mu$g m$^{-3}$) dust mass concentrations in Miami is a less direct transport of dust, with the back-trajectories passing over the Gulf of Mexico and then recirculating back to Miami via the lower southeast United States. This is most evident in 2014 and 2015. This transport pathway is effective at dispersing dust over large areas of southeastern United States, and increase the likelihood of dust interacting with other terrestrial and anthropogenic aerosols en route.

## 4   Comparison of GEOS-5 FP/MERRA-2 aerosol distributions to the Miami observations

### 4.1   Aerosol optical depth

The GEOS-5 FP $\tau_{aer}$ values slightly exceed those from the sun photometers at optical depths below 0.15, and are slightly less at $\tau_{aer} > 0.15$ (Fig. 5). The slight underestimation at higher $\tau_{aer}$s is also shown in Shi et al. (2019), in which it is attributed to missing emissions (Buchard et al., 2017), and is also apparent in a comparison of MERRA-2 $\tau_{aer}$s to shipborne $\tau_{aer}$ observations (Fig. 9 of Randles et al., 2017). Surface albedo inhomogeneities at this coastal location could perhaps contribute to the

overestimate at the lowest $\tau_{aer}$s, although another interpretation, shown next, is that an overestimate from the sea salt contribution becomes most apparent at the lowest $\tau_{aer}$s. An implication of the positive correlation, overall, is that the AERONET cloud screening is effective. Although not shown, the day-to-day variations in $\tau_{aer}$ match well, as would be expected given the assimilation and availability of satellite-derived $\tau_{aer}$s.

### 4.2   Surface sea salt mass concentrations

A first step within the assimilation scheme is the aerosol speciation, and how much of the aerosol is partitioned into sea salt will also affect the amount available for dust. The GEOS-5 FP aerosol product overestimates the near-surface salt mass con-



centration by an order of magnitude, with a model mean value of 61 $\mu$g m$^{-3}$ versus a measured mean salt mass concentration of 7.7 $\mu$g m$^{-3}$ estimated from the sodium measurements (Fig. 6). The latter is consistent with previous measurements (Savoie and Prospero, 1982; Prospero, 1999a). Further assessment of the GEOS-5 wind speeds and relative humidity using surface meteorological data indicate that these are realistic (not shown), suggesting that the underlying size and/or light extinction pa-

5 rameterizations may be the issue. This is consistent with the findings of Bian et al. (2019), in which an inconsistency between an overpredicted salt mass concentration and underpredicted $\tau_{seasalt}$ in GOES-5 simulations is reconciled through the model adoption of too-large sea salt sizes. As noted by Buchard et al. (2017), the assimilation of $\tau_{aer}$s that exceed the forecast $\tau_{aer}$s can further exacerbate the discrepany.

## 4.3 Surface dust mass concentrations

All else equal, an overestimate of the GEOS-5 FP sea salt mass concentrations might point to GEOS-5 FP dust mass concentrations that are too low, for the same assimulated aerosol optical depth. Instead, the GEOS-5 FP near-surface dust mass concentrations also exceed observed values on almost two-thirds of the days, although the overestimation is not as pronounced as that for sea salt, with a mean bias of 3.46 $\mu$g m$^{-3}$ (Fig. 7). A clear correlation is apparent, matching that reported in Buchard et al. (2017) for MERRA-2 based on Barbados measurements (which did not find a systematic bias). GEOS-5 FP approxi-

mately captures the monthly evolution from July to September, although the GEOS-5 FP overestimate is most pronounced for June, and is enough to alter the perception of the monthly evolution (i.e. the GEOS-5 FP maximum in 2014 and 2016 occurs in June, in contrast to a July (August) maximum in the observations). The cause for this is unclear.

The model overestimate in the surface dust mass concentrations has consequences for inferences of the frequency of high and low dust loading days (Fig. 8). Measured dust mass concentrations exceeding 40 $\mu$g m$^{-3}$ or less than 1 $\mu$g m$^{-3}$ were

20 infrequent, with dust present in the Miami boundary layer at concentrations between 1 to 10 $\mu$g m$^{-3}$ over one-half of the time. In contrast, GEOS-5 FP include more days with concentrations exceeding 10 $\mu$g m$^{-3}$ than are observed, and correspondingly fewer days with concentrations between 1-10 $\mu$g m$^{-3}$. GEOS-5 FP overall places too much dust in the boundary layer. That dust is consistently present in the boundary layer is not controversial (e.g., Reid et al., 2002).

## 4.4 Does dust size explain the difference between the GEOS-5 FP/MERRA-2 and measured dust mass

concentrations?

The use of a size distribution parameterization that permits a larger mass for the same visible extinction (as is thought to occur for sea salt; Bian et al. (2019)) is one explanation for why the MERRA-2 dust mass concentrations may exceed those observed, despite realistic assimilation-constrained aerosol optical depths. This idea is assessed using size-resolved dust mass concentrations from 17 days in 2016, collected in addition to the bulk filter samples. The 28 July to 9 August time period

included one of the heaviest and longest dust mass concentration events from 2016, and data from another dust event spanning 1-2 September are also included. The trajectories for these days indicate the dust transport occurred directly over the Atlantic to Miami without passing over the Gulf of Mexico and recirculating back. By happenstance, the subtropical high was strongest and located furthest north in 2016 of the three years considered, based on the 1023 hPa sea level pressure contour (Fig. 4).





This may have decreased dust transport that year (Kramer et al., 2019), resulting in relatively fewer days with dust exceeding 20 $\mu$g m$^{-3}$ for that year. The episodic-maximum dust mass concentration of 28 $\mu$g m$^{-3}$ occurred on August 4, 2016 (Fig. 2). The days with the largest size particles (7/30, 8/2, and 8/4) correspond to high lidar volume depolarization ratios and total bulk surface dust mass concentrations (Fig. 3).

Overall, the GEOS-5 FP size distribution is broader than that measured, with GEOS-5 FP placing 16% of the dust mass concentration in diameters < 2 $\mu$m, with 46%, 32%, 5.4% and 0.05% in the subsequent larger size bins. In contrast, the measurements place 56% of the total dust mass concentrations in particles with diameters < 2.1 $\mu$m, with 18%, 21%, and 3.6% in the larger size bins. We note that the model size distribution places equal mass in the three size bins with diameters exceeding 2 $\mu$m, and that thereafter, the only mechanism that can alter the dust sizes is differential settling. This effect is

evident in the GEOS-5 dust size distribution after long range transport, with most of the mass contained in the (2.0 - 3.6 $\mu$m) diameter range, and the mass distribution within the three larger size bins no longer equal. As has been previously noted by others (e.g., Kok, 2011), more of the measured mass is contained in particles with diameters > 6 $\mu$m, than in the model. Filters from 1 and 2 August indicate mass from particles with diameters > 10 $\mu$m, and these were interrogated further with scanning electron microscopy to verify the particle size (Fig. 10). The dust particle images independently corroborate the presence of

larger sizes. Four of the five examples have a dimension exceeding 20 $\mu$m (see Table 2 for dimensions of each identified particle), with the largest particle measuring 36.65 by 26.86 $\mu$m (Fig. 10b). The particles are highly aspherical, with aspect ratios ranging from 0.38 to 0.73. This could contribute to their survival, as aspherical particles will fall at slower terminal speeds than spheres of equivalent mass (e.g., Yang et al., 2013). These dust particles exceed the GEOS-5 upper limit of a 20 $\mu$m diameter. However, only 1 $\mu$g m$^{-3}$ of the total measured dust mass concentration is contained in particles with diameters

exceeding 10 $\mu$m, implying that the neglect of particles with diameters > 20 $\mu$m by GEOS-5 FP only results in a small error in the total dust mass concentration after such long-range transport.

The more striking result is that the measurements place more than half of the mass in particles with diameters < 2 $\mu$m, or more than three times as much as does GEOS-5 FP. Particles with diameters < 2 $\mu$m possess larger mass extinction efficiencies than larger particles (Table 1), by at least a factor of three, implying that more mass is required to produce the same extinction

for larger particles than for smaller particles. Fig. 9 indicates that one cause for the overestimated near-surace dust mass concentrations in GEOS-5 FP above southern Florida may be a dust size parameterization that distributes most of the dust mass into a larger size, for which the mass extinction efficiency is lower.

We are not aware of other dust size distribution measurements gathered over coastal Florida against which to compare this study's measurements. Measurements made close to the African coast emphasize the presence of larger particles, with Ryder

et al. (2013) reporting most of their measured dust mass in diameters exceeding 5 $\mu$m, consistent with Haywood et al. (2003). Kok et al. (2017) show that most of the dust mass at emission is contained in particles with diameters reaching 10 $\mu$m. The largest particles sediment out during transport, and *in-situ* measurements made during the Puerto Rico Dust Experiment most of the mass is in particles with diameters between 5-7 $\mu$m (Maring et al., 2003b; Reid et al., 2003b), with similar sizes between the boundary layer and the free troposphere (Maring et al., 2003a). More of the dust mass is reported in larger sizes than

is reported here for south Florida. In contrast, Inferences of dust size deduced from multi-wavelength lidar measurements at





Barbados (Haarig et al., 2017) estimate most dust particles are 1.6-2.0 $\mu$m in diameter, which is more broadly consistent with the Miami impactor measurements. This is at first glance consistent with the *in-situ* dust mass concentrations reported in Jung et al. (2013), but these do not extend beyond 2.5 $\mu$m in diameter (most aircraft aerosol intake inlets cut off at 3 $\mu$m), thus do not resolve the larger sizes.

## 4.5  Dust vertical structure

The GEOS-5 FP dust mass mixing profile combined with the mass extinction efficiencies specific to each size range (Table 1) can be assessed using extinctions retrieved from the lidar backscattered intensities (Delgadillo et al., 2018). This is similar to the strategy invoked within Liu et al. (2012) for comparison between space-based lidar extinctions and GEOS-5 dust mass loadings, usingh a mass extinction efficiency of 0.7 m$^2$ g$^{-1}$, and Sauter et al. (2019) (who use a smaller value of 0.4 m$^2$ g$^{-1}$, implying larger particles). Four examples, spanning the 28-30 July 2016 dust event and that of 5 August, 2016, are shown in Fig. 11. The AERONET-derived $\tau_{aer}$s from the early morning, overlapping with 12 UTC, indicate total $\tau_{aer}$s of 0.1-0.2, with the highest aerosol loading occurring on 29 July, 2016.

The time series of the lidar-derived extinctions (Fig. 11c,f,i and l, all corresponding to nighttime, when the lidar signal is more robust) indicate dust extending up to 4 km at times, stratified into distinct layers supporting what appear to be gravity waves (Fig. 11). A peak at 2 km is specific to the 28-30 July dust event (Fig. 3), and a secondary peak at approximately 800 m represents the boundary layer top, where the maximum in relative humidity may induce the hygroscopic swelling of sea salt (or of other hygroscopic aerosols mixed with dust), or indicate undetected optically-thin cloud (Delgadillo et al., 2018).

The GEOS-5 FP dust mass concentrations clearly indicate the presence of dust in the free troposphere on two of the days, though, perhaps surprisingly, not on the day with the highest AERONET-derived optical depth. When dust is present, its vertical distribution is similar to that inferred from the lidar. Within the boundary layer, the filter-based size-resolved dust mass concentrations measurements (thick red bars within Fig 11a,d,g and j) reach 10 $\mu$g m$^{-3}$ or more on all four days. GEOS-5 FP near-surface dust mass concentration values are higher on all four days than those measured (compare to the thick black bars).

Corresponding model aerosol extinction profiles, resolved over the 3-12 UTC time frame encapsulated within the average of the cloud-free lidar extinction profiles, indicate that the GEOS-5 FP product does not distribute enough of the assimilated $\tau_{aer}$s above 1.5 km on any of the four days (Fig. 11b,e,h,k). Most of the model $\tau_{aer}$ is confined to the boundary layer, where the $\tau_{aer}$ also contains a substantial sea salt contribution given relative humidities capable of exceeding 80% (RH not shown). The lidar extinction values within the boundary layer are much lower than those from GEOS-5 FP, and, are lower than those in the free-troposphere, despite including a sea salt contribution. Overall this assessment suggests that GEOS-5 FP places too much dust within the boundary layer, and not enough in the free troposphere above the boundary layer, where the dust can be advected further more easily. Where GEOS-5 FPl does place dust within the free troposphere, its vertical distribution is reasonable, however.

Agreement between the model-derived and observed lidar extinctions, for the same dust mass concentrations, can in theory be produced through the application of higher mass extinction efficiencies. The model tendency to overestimate the mean particle size could also be one reason why the model may place too much of the dust at lower altitudes (Fig. 9). Since larger





particles fall more quickly, they should be preferentially located at lower altitudes. Size-resolved model dust mass mixing ratios for 28 July 2016 indicate that larger particles do prefer lower altitudes (Fig. 12), and in particular much fewer particles with diameters $> 6$ $\mu$m occur above the boundary layer. In the two size ranges with the largest discrepancy between the model and measurements (2-3.6 $\mu$m versus 0.2-2 $\mu$m), the altitude difference between the vertical distribution of the two size-resolved

dust mass mixing ratios is only on the order of 100-300 m, however.

## 5 Conclusions

Dust forecasts incorporating the assimilation of satellite-derived $\tau_{aer}$ can circumvent a difficulty global aerosol models otherwise encounter, in which $\tau_{aer}$s tend to be too low further away from a source region (e.g., Kim et al., 2014; Evan et al., 2014; Ansmann et al., 2017). As such, the assimilation of observed $\tau_{aer}$ holds the promise of more accurate depictions of

10 the global aerosol distribution. While the GEOS-5 FP aerosol forecasts and MERRA-2 capture independently-measured $\tau_{aer}$s and their variability well for an aerosol environment dominated by sea salt and dust transported far from its source region, the more challenging objectives of realistic aerosol vertical distributions and surface mass concentrations are less well met. A clear overestimate in the modeled sea salt loading, apparent in an *in situ* comparison (Fig. 6) and a comparison of model-derived extinctions to those from a lidar (Fig. 11), is corroborated by the independent findings of Bian et al. (2019), will foster an

15 underestimate in the dust loading, for the same assimilated $\tau_{aer}$.

A comparison to dust mass concentrations measured at the surface and to lidar profiles of retrieved extinction indicates that GEOS-5 FP often distributes dust too low in the atmosphere, with too much mass placed in particle sizes that are larger than observed (except at the largest sizes). An overestimate of the amount of dust in the boundary layer has implications for model-deduced ocean fertilization by the soluble iron (e.g. Colarco et al., 2003), and for cloud nucleation. Size-resolved measurements

place most of the mass in diameters smaller than 2 $\mu$m, while GEOS-5 FP/MERRA-2 places most of the dust mass in diameters between 2 to 3.6 $\mu$m. The prescribed dust mass extinction efficiency, by which the dust portion of the assimilated aerosol optical depth is cast as a dust mass mixing ratio, is more than a factor of 3 larger for the smaller size (Table 1). Thus, a model dust size distribution that places more of the dust in the smaller sizes, but otherwise using the same prescribed size-resolved mass extinction efficiencies, will place less dust mass within the boundary layer. A relative increase in the amount of dust within the

smaller sizes in the free troposphere may be important for the overall large-scale spatial distribution of the dust, although the vertical distribution of dust differs little for the smaller sizes (Fig. 12).

The overestimate in the model dust size after long-range transport is opposite to that documented for most global aerosol models, in which the number of small particles can be overestimated relative to the large particles (e.g., Kok, 2011). We recognize a recent emphasis on the presence of very large dust particles within the Saharan Air Layer (e.g., Ryder et al., 2019),

and their ability to be transported for long distances. Large aspherical dust particles are also detected at Miami after a transport of at least 20,000 km, exceeding the amount modeled, but their contribution to the overall mass is negligible at 1 $\mu$g m$^{-3}$. Their presence is consistent with other observations that perceive little mixing of African dust with other air masses en route for select cases (Karyampudi et al., 1999), and find larger dust particles closer to the top of the Saharan air layer (Jung et al.,



2013; Yang et al., 2013; Gasteiger et al., 2017). The existence of the large particles may be more typical of the dust events that advect more directly to Miami and undergo little precipitation. The neglect of particles exceeding the maximum GEOS-5-specified particle diameter of 20 $\mu$m introduces little error in the total GEOS-5 dust mass concentrations, but may be more important for the direct aerosol radiative effect, which is not addressed here. Overall global aerosol models with and without
aerosol assimilation benefit from a more realistic modeling of dust particle size and its evolution with transport and age after emission (Adebiyi et al., 2019).

Our dust size measurements, at a location further away from the dust source than Puerto Rico (Reid et al., 2002, 2003b; Maring et al., 2003b, a) and Barbados (Jung et al., 2013; Weinzierl et al., 2016), are difficult to compare in a consistent manner to those reported in previous studies. *In-situ* measurements made during the Puerto Rico Experiment suggest most of the dust
mass resides in diameters between 5-7 $\mu$m, while recent lidar measurements tend to perceive larger concentrations of aerosols with diameters $< 2$ $\mu$m (Haarig et al., 2019). Reasons for such discrepancies are not entirely understood. Neither the impactor measurements nor the GOCART dust size parameterization discriminates for aerosol diameters $< 2$ $\mu$m, and published dust size distributions are not always directly comparable. Although critiques can be made of each individual measurement presented within this study, in their totality a consistent interpretation emerges based on retrieved lidar extinctions, and near-surface dust
and sea salt mass concentrations, suggesting the model mean dust sizes, by mass, are too large, with relatively too much dust mass placed in the boundary layer. The total dust loading is too low within GEOS-5 FP and MERRA-2, possibly because too much of the assimilated aerosol optical depth is speciated into sea salt. We consider the current analysis a pilot study, however, and recommend a further dedicated assessment with a multi-wavelength depolarization extinction lidar as well as a more complete set of *in-situ* measurements, both at the ground and in the free troposphere. This will better anchor ideas for
changes to the GEOS-5 sea salt and dust size parameterizations.

*Data availability.*   The dust mass concentration dataset is available at the University of Miami Data Repository at https://doi.org/10.17604/q3vf-8m31, for which Zuidema et al. (2019) is the appropriate reference. The datasets developed for this study are available from the first author (lidar extinction retrievals, size-resolved dust mass concentrations and measured sodium mass concentrations), and the sun photometer aerosol optical depths are publicly available through the AERONET website.

*Author contributions.*   PZ designed the study and led the writing of the manuscript. SK carried out the data collection and initial analysis at Miami and drafted the initial manuscript. The GEOS-5 FP and MERRA2 values were provided by RG. RD provided the lidar analysis. CA contributed to the Miami data collection, with LC providing input on the methodology and additional mentorship of SK. AB provided the SEM analysis. PC provided substantial insight into the GEOS-5 and MERRA2 methodology. All authors commented on the manuscript.

*Competing interests.*   No competing interests are present





*Acknowledgements.* We gratefully acknowledge support for the acquisition of the micropulse lidar and sun photometers through an NSF Major Research Instrumentation grant 0923217. PZ, SK and CA acknowledge additional support from a Faculty Program Fund, NSF AGS-1233874 and an NSF AGS Research Experiences for Undergraduates grant. We thank Arlindo da Silva for intellectual input on the original study. RD acknowledges support from DOE ASR grant DE-SC0013720. We thank Brent Holben and Tom Eck for their support of the Miami sun photometers through the AERONET project. This work would not have been possible without the history of investments by Dr. Joseph Prosprero into the dust filter sampling program at U of Miami, its analysis and intellectual interpretation.



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





**Table 1.** Particle diameter range applied within GEOS-5 dust size parameterization, corresponding mass extinction efficiencies (MEE) at 550 nm wavelength based on the non-spherical optics model presented in Colarco et al. (2014), and Miami impactor size selections.

| density | GEOS-5 FP | MEE | Miami |
|---|---|---|---|
| kg m$^{-3}$ | $\mu$m | m$^2$ g$^{-1}$ | $\mu$m |
| 2500 | 0.2-2.0 | 2.02 | < 2.1 |
| 2650 | 2.0-3.6 | 0.64 | 2.1-4.2 |
| 2650 | 3.6-6.0 | 0.33 | 4.2-10.2 |
| 2650 | 6.0-12.0 | 0.17 | >10.2 |
| 2650 | 12.0-20.0 | 0.08 | NA[1] |

[1] not available





**Table 2.** Measured size of imaged particles in Fig. 9.

| particle[1] | length | width | length/width |
|---|---|---|---|
|  |  | $\mu$m | $\mu$m |
| a 1 | 26.8 | 11.5 | 0.43 |
| a 2 | 17.0 | 12.0 | 0.70 |
| b | 36.6 | 26.9 | 0.73 |
| c | 27.8 | 14.6 | 0.52 |
| d | 21.6 | 8.3 | 0.38 |

[1] indicated by panel label and image number



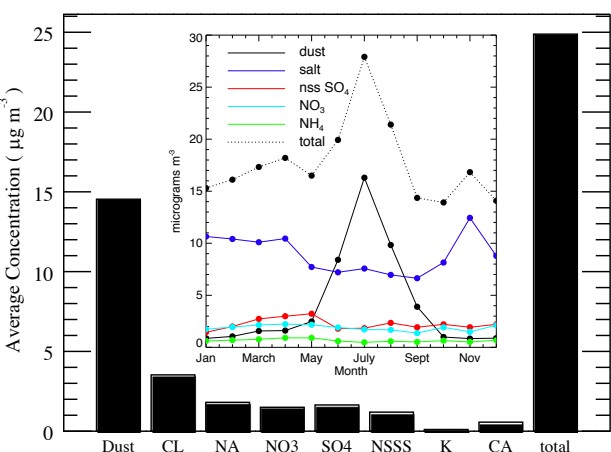

**Figure 1.** Average aerosol concentrations for July - August 2014 of the soluble ions chlorine (CL), sodium (NA), nitrate (NO3), sulfate (SO4), and non-sea-salt sulfate (NSSS), potassium (K), and calcium (CA). Inset plot is the Miami monthly-mean aerosol concentration by species from 1989 through 1996, replotted from Prospero (1999a).

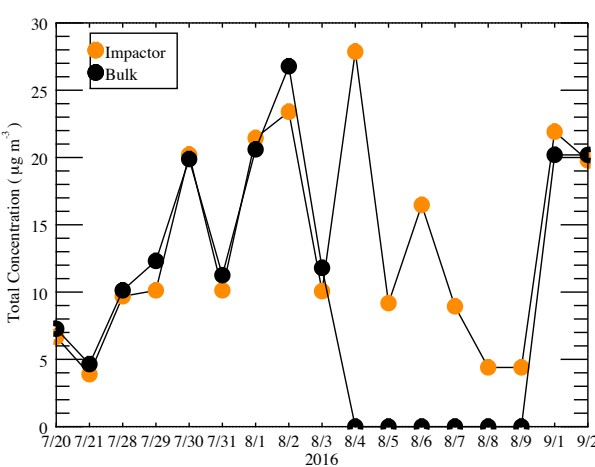

**Figure 2.** Daily bulk filter dust mass concentrations (black filled circles) and the sum of the impactor filter mass concentrations (orange filled circles) for 20 July - 2 September, 2016. No bulk mass concentration data are available for 4-9 August.



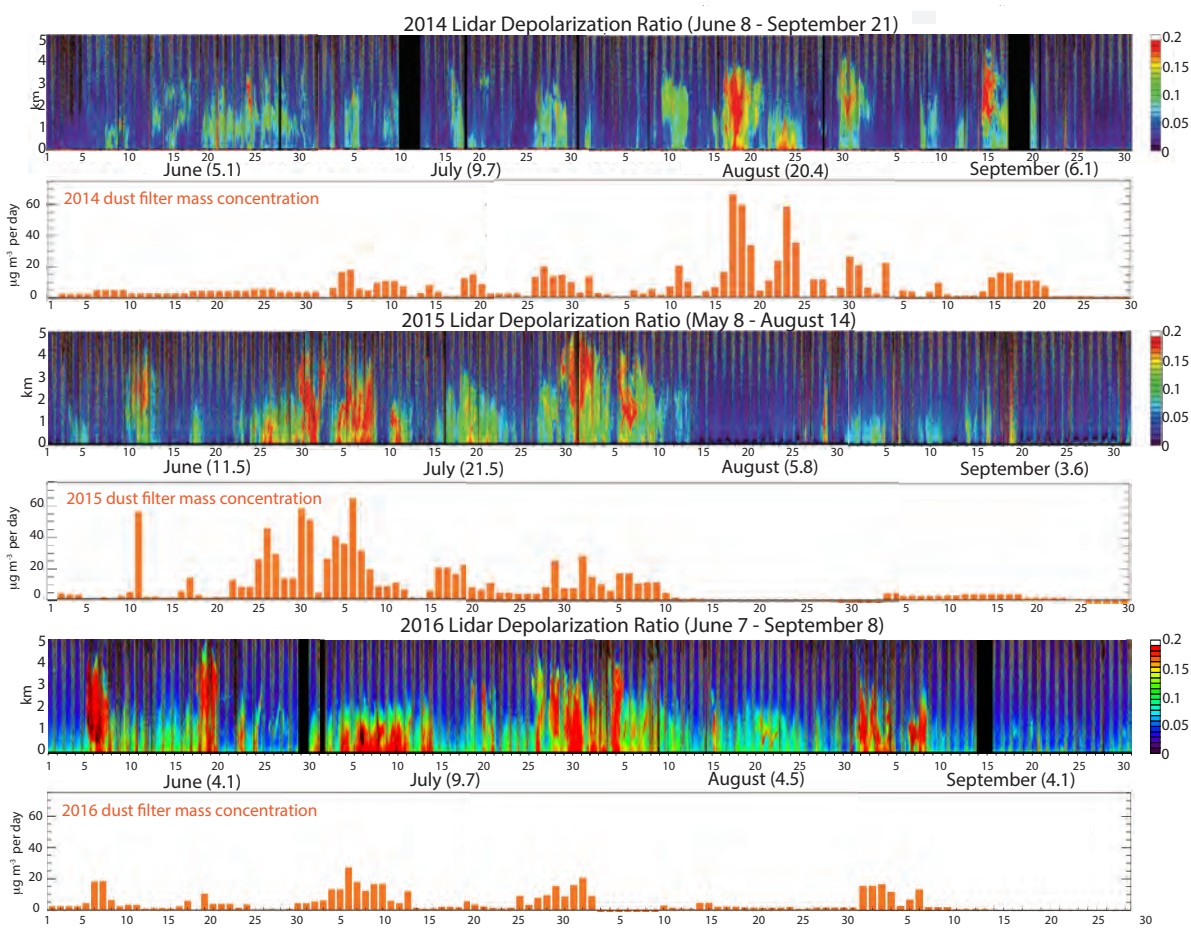

**Figure 3.** Time series of the MPL volume linear depolarization ratios for June-September, 2014, 2015, 2016, interspersed with the corresponding time series of the daily filter-retrieved bulk dust mass concentrations. Subjectively-determined dates of the first and last days with significant dust is indicated within the lidar panel titles. Monthly-mean surface dust mass concentrations are indicated for each month.



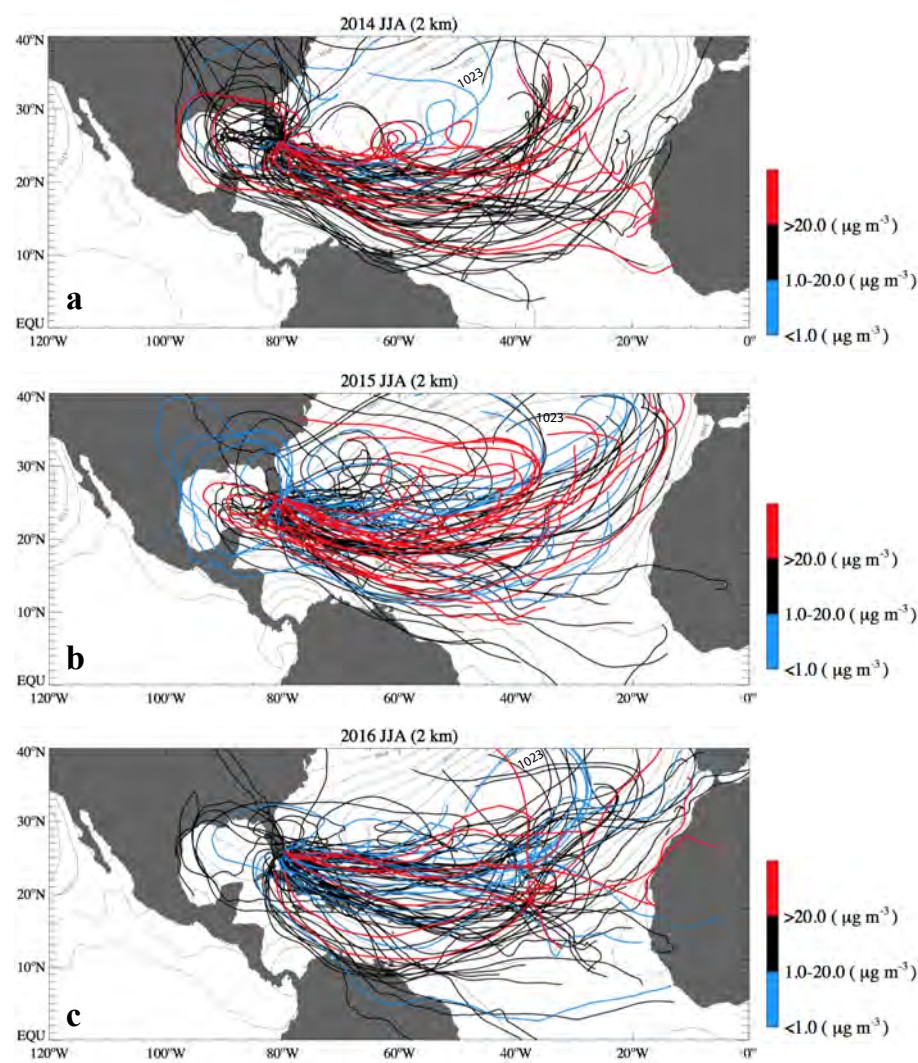

**Figure 4.** Daily HYSPLIT back trajectories for June-August of a) 2014, b) 2015, and c) 2016, extending for 300 hours and initialized at a height of two km above sea level, with the average seasonal NCEP Reanalysis sea level pressure (grey contours). Back trajectories are color-coded by the dust mass concentrations measured at Miami: initially exceeding $> 20$ $\mu$g m$^{-3}$ (red), 1 - 20 $\mu$g m$^{-3}$ (dark blue), and $< 1$ $\mu$g m$^{-3}$ (light blue).



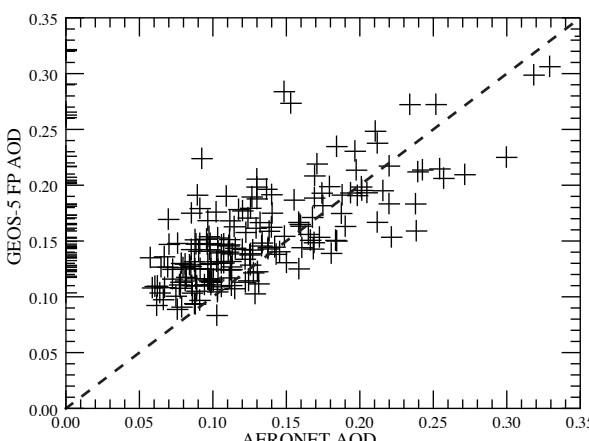

**Figure 5.** Daily-mean GEOS-5 FP aerosol optical depths versus AERONET-derived values at 500 nm wavelength, for June-September 2015 and 2016.



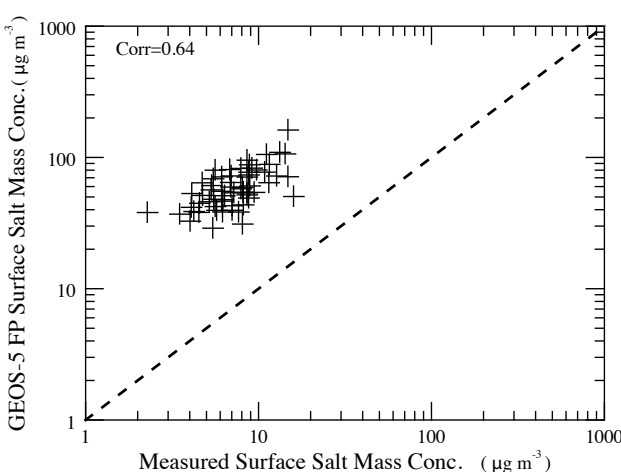

**Figure 6.** Daily-mean GEOS-5 FP versus measured near-surface salt mass concentrations for July-August, 2014 (32 days). The measured sea salt concentration is computed from the sodium concentration multiplied by 3.256, following Prospero (1999a).



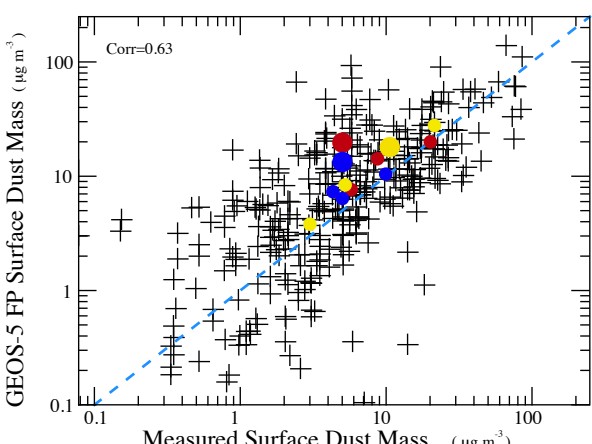

**Figure 7.** GEOS-5 FP daily versus measured near-surface dust mass concentrations for June-September from 2014-2016 (366 days), with monthly-mean values superimposed for 2014 (dark red), 2015 (dark yellow) and 2016 (blue); a larger filled circle highlights June. y=x line indicated. Correlation of 0.63.

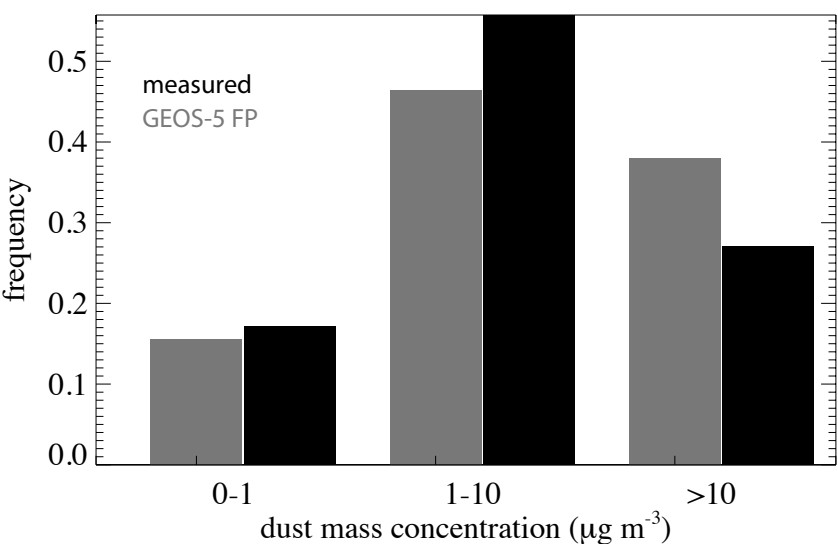

**Figure 8.** Frequency of days with measured (black) and model (grey) daily-mean dust mass concentrations in three diamater bins, based on 366 days from June-September, 2014-2016.



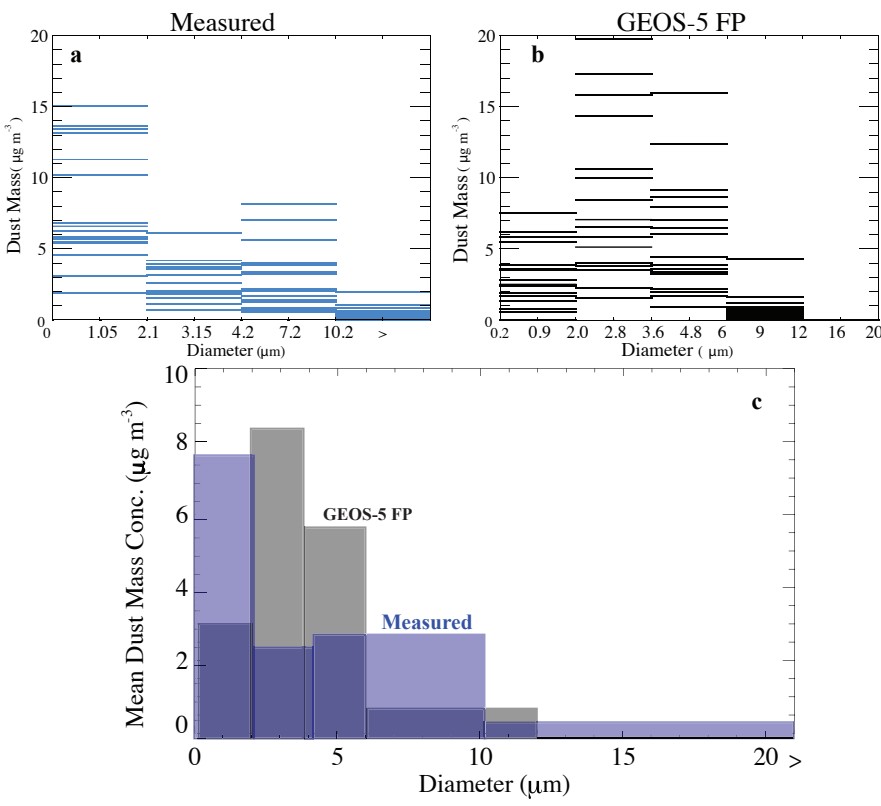

**Figure 9.** a) Daily-mean dust mass concentration over each diameter bin for each of the 17 sample days in measurements and b) GEOS-5 FP (black). c) 17-day-average dust mass concentration for each diameter bin from measurements (blue) and GEOS-5 FP (black). Note that GEOS-5 FP and the measurements fall within separate diameter bins (see Table 1).





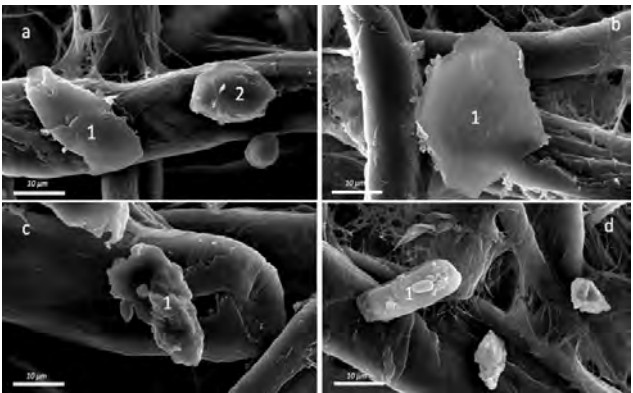

**Figure 10.** Scanning electron microscopy images of filter samples from a) and b) August 1, 2016, and c) and d) August 2, 2016. Individual particle dimensions are shown in Table 2.



**Figure 11.** a) MERRA-2 dust (solid) and sea salt (dashed) mass concentrations for 28 July 2016 3-12 UTC (dark blue, light blue, green and yellow lines indicate 3, 6, 9, and 12 UTC respectively), with surface model (black) and impactor (red) total dust mass concentrations included as thick vertical lines. b) GEOS-5 FP dust (solid) and total (dashed) volume extinction coefficients for 28 July 2016 3-12 UTC (same labeling scheme as in a)) and the lidar-derived volume extinction coefficient profile averaged over cloud-free portions (determined visually from camera imagery) of 28 July 2-11 UTC (black). The average 11-13 UTC AERONET-derived aerosol optical depth at 500 nm is indicated. c) 28 July 2011 UTC lidar-derived volume extinction coefficient time series. d)-f) same as a)-c) but for 29 July 2016. g)-i) same as a)-c) but for 30 July 2016. j)-l) same as a)-c) but for 5 August 2016. 0-12 UTC corresponds to 19-7 local time.



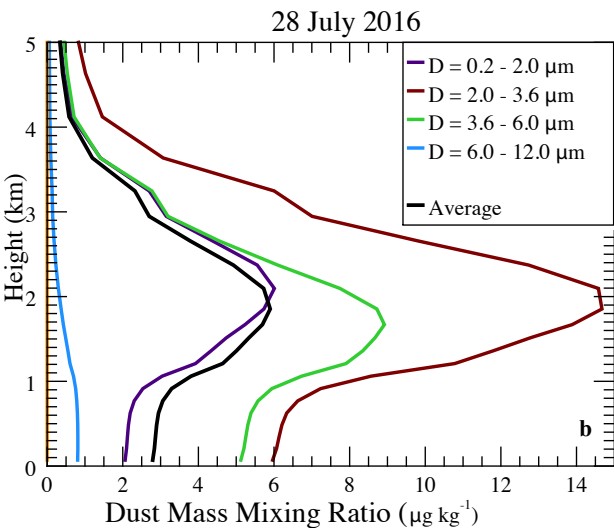

**Figure 12.** Size-resolved GEOS-5 FP dust mass mixing ratio vertical profiles for 28 July, 2016.