# Peer review of "Apparent dust size discrepancy in aerosol reanalysis in north African dust after long-range transport"

_Atmospheric Chemistry and Physics, 2020_

## Referee Comment (RC1) · Jasper Kok (Referee) · 24 Apr 2020

Review of Kramer et al. for ACP This paper investigates the size distribution of long-range transported African dust at Miami, Florida and compares in situ measurements of the (size-resolved) concentration against aerosol reanalysis data. They bring to bear a wide variety of experimental and numerical tools, and the breadth of the analysis is impressive, making for overall robust results. The authors find that the surface concentration of dust at Miami is somewhat overestimated by the reanalysis products and is also finer than represented in these (highly similar) reanalysis products. This last conclusion is counter to findings by several other studies, including my own, so is

surprising and possibly important.

The article is well written and is overall excellent science. I do have some comments the authors should address before publication:

 c The conclusion regarding the overestimation of dust surface concentration by the reanalysis products is confusing in the abstract. Line 7 of the abstracts notes: "Measured near-surface dust mass concentrations slightly exceed model values" whereas line 13 notes: "model dust mass concentrations near the surface are higher than those measured". Please resolve these seemingly contradictory statements.

 c A methodological problem is that the authors are comparing modeled dust size distributions in terms of geometric (volume-equivalent) diameter $D_g$ against measured aerodynamic diameters $D_a$. But because dust is both very aspherical (e.g., Okada et al., 2001; Kandler et al., 2007; Huang et al., 2020) and has a much larger density than water, a particle's geometric diameter is smaller than its aerodynamic diameter and a correction should be made. I recommend using the simple correction based on shape data in Kok et al. (2014), namely $D_g = 0.75 D_a$.

 c I'm confused about GEOS-5 FP versus MERRA-2. The abstract describe these as "closely related" but section 2.4 ("GEOS-5 FP/MERRA-2") only includes a description of GEOS-5 FP and does not discuss MERRA-2. Please clarify the distinction (if any?) between these two products.

 c Data availability – the authors note data is available from the first author, but these data are valuable to the community and really ought to be posted on a publicly available repository.

 c I think Figure 9 is the paper's most salient result – that surface dust in Miami is finer than represented in aerosol reanalysis products – but some corrections need to be made to the presentation. Panel c shows dust mass concentration versus diameter in terms of bars, and a reader would reasonably conclude that the surface area of the bar

is representative of the mass of aerosol in that size class. But that is not the case here, and instead the total mass in each size class is plotted. This leads to a distortion of the data in that larger size bins will be correspondingly larger (e.g., if you divide the 4-10 um bin into two separate bins, then each bin would be only half the height of the current representation). To fix this, the authors need to normalize the mass concentration by the diameter range, which is equivalent to plotting dM/dD, as is standard (although they could also plot dM/dlnD and use a logarithmic x-axis).

Minor comments: • Line 27-29 on p. 4: "some reduction in aerosol asphericity may also occur from chemical aging". Measurements indicate that chemical aging of North African dust is quite limited (Denjean et al., 2015), and Huang et al. (2020) just showed that North African dust becomes more aspherical during transport, probably because of preferential settling of spherical dust particles. So this statement should be adjusted.

• P. 11: "transport of at least 20,000 km". Earth's circumference is ∼40,000 km, so this seems incorrect.

• Second column of Table 1 should specify that this is diameter.

• Table 2. Length > width and aspect ratio >= 1, so the entries in the last column (which are all < 1) should be the reciprocal number. It'd be interesting to note how the measured aspect ratios compare to other literature data on dust shape, as compiled in Huang et al. (2020).

• Figure 2. If there is no data, then no data should be plotted, so please remove the zeroes for the bulk mass concentration data (8/4-8/9).

• Figure 3: please note units for dust mass concentrations in caption.

• Figure 7: please note whether correlation is taken in linear or logarithmic space (it ought to be the latter since the data spans several orders of magnitude).

References

Denjean, C., Caquineau, S., Desboeufs, K., Laurent, B., Maille, M., Rosado, M.Q., Vallejo, P., Mayol-Bracero, O.L., Formenti, P., 2015. Long-range transport across the Atlantic in summertime does not enhance the hygroscopicity of African mineral dust. Geophys. Res. Lett. 42, 7835-7843.

Huang, Y., Kok, J.F., Kandler, K., Lindqvist, H., Nousiainen, T., Sakai, T., Adebiyi, A., Jokinen, O., 2020. Climate Models and Remote Sensing Retrievals Neglect Substantial Desert Dust Asphericity. Geophys. Res. Lett. 47, e2019GL086592.

Kandler, K., Benker, N., Bundke, U., Cuevas, E., Ebert, M., Knippertz, P., Rodriguez, S., Schuetz, L., Weinbruch, S., 2007. Chemical composition and complex refractive index of Saharan Mineral Dust at Izana, Tenerife (Spain) derived by electron microscopy. Atmos. Environ. 41, 8058-8074.

Kok, J.F., Mahowald, N.M., Fratini, G., Gillies, J.A., Ishizuka, M., Leys, J.F., Mikami, M., Park, M.S., Park, S.U., Van Pelt, R.S., Zobeck, T.M., 2014. An improved dust emission model - Part 1: Model description and comparison against measurements. Atmos. Chem. Phys. 14, 13023-13041.

Okada, K., Heintzenberg, J., Kai, K.J., Qin, Y., 2001. Shape of atmospheric mineral particles collected in three Chinese arid-regions. Geophys. Res. Lett. 28, 3123-3126.

---

## Referee Comment (RC2) · Anonymous Referee #2 · 14 May 2020

Review of manuscript acp-2020-1

GENERAL REMARKS

The study investigates the size distributions of mineral dust from North Africa after long-range transport across the Atlantic to Miami, Florida. Data stem from remote sensing instrumentation (MODIS, AERONET and MPL), in-situ data from impactor samples, reanalysis data (MERRA-2) and forward processing product GEOS-5 FP, and span over three dust seasons from June to August in 2014, 2015, and 2016.

The key finding of the study is that the surface mass concentrations of mineral dust provided by the reanalysis products overestimate the values from surface sampling slightly. Furthermore, the dust represented in the models is of finer granularity than reported from in-situ observations. Overall, the presented results suggest that model mean dust sizes by mass are too large with too much dust mass placed in the boundary layer. These conclusions are of interest for the scientific community and deserve publication in ACP.

The manuscript is clearly structured and in general terms well written. In few sections it lacks quantification of results, which requires modification. The manuscript can be accepted for publication in ACP after major revisions have been considered which are specified in the following. Revisions are classified major since the adjustment of diameters to either volume equivalent or aerodynamic equivalent may alter the main conclusions.

SPECIFIC COMMENTS

1. The abstract contains the conclusion "Measured near-surface dust mass concentrations slightly exceed model values, with most of the modelled dust mass in diameters between 2-6 μm." The data shown in Fig. 6 show the opposite behaviour. Please clarify.

2. The results presented graphically in Figs. 5 -7 are described in the text only qualitatively as "robust correlation" (page 4, line 13), "slight underestimation" (page 7, line 22) or "clear correlation" (page 8, line 13). In all cases the results from a linear correlation analysis should be presented and discussed in the text.

3. The presentation and discussion of dust particle sizes lacks clarity. Size distributions where measured by a cascade impactor which sorts particles according to their Stokes number and thus aerodynamic equivalent diameter which assumes unit density (1 g cm$^{-3}$). The models use spheroidal particles to describe dust particles with given densities. All values are listed in Table 1, but there is a conversion missing from aerodynamic equivalent diameter with density 1 g cm$^{-3}$ volume equivalent diameter with densities 2.5 to 2.65 g cm$^{-3}$. However, this conversion is essential for the comparison pf the results and needs to be included. Furthermore, the title of column 4 should be modified from "Miami" to, e.g., "Miami impactor cut-off diameter".

Figure 9 showing the results form the size distribution comparison need to be adjusted to the respective size classes after the diameters have been converted to one type (either volume

equivalent or aerodynamic equivalent). Besides adjusting the diameters of one data set to the other, no matter which serves as reference, the size distributions need to be presented as dM/dlnD or dM/dlog D, or dM/dD, whatever is preferred. Otherwise the size distributions cannot be compared.

Finally, Fig. 12 lacks explanation. To my understanding, it shows the vertical profiles of mass mixing ratios for the 5 size classes of GEOS-5 FP, and thus the vertical distribution of the total dust mass mixing ratio in the model would be the sum of the five classes. If this is correct, what is then the "Average"? Here, some explanation is requested.

MINOR ISSUES

Page 2, line 32: blank space between "a" and "MPL".

Page 4, line 31: It might be easier to read if the equations are presented on a separate line.

Page 6, line 25: The Chapter title "Overview" is misleading. The title should contain an indication that here the presentation of results starts, e.g., "Overview of dust seasons from 2014 to 2016".

Section 4.4: Figure 10 seems to be introduced before Figure 9, please check.

Page 9, line 35: use lower –case letter after comma.

Page 11, line 8: the term "$\tau_{aer}$s" is confusing, since it also may refer to the product of aerosol optical depth $\tau_{aer}$ and a property $s$. Please rephrase.

Page 22, Figure 2: please remove the points from the plot where no data are available. The current plot is heavily misleading.

Page 28, Figure 8: it would be of high relevance to the variability range of the average frequencies connected to the respective mass concentration bins.  Please show -±1$\sigma$ as error bars.

---

## Author Comment (AC1) · 6 Jun 2020

Response to the reviews for Kramer et al., "Apparent dust size discrepancy in aerosol reanalysis in north African dust after long-range transport", acp-2020-1

We would like to thank Jasper Kok and anonymous referee #2 for taking the time and focus to provide their thoughtful reviews during what has been a difficult spring for everyone. These reviews are greatly appreciated. We include each one below, with responses provided in a blue font.

Jasper Kok:

This paper investigates the size distribution of longrange transported African dust at Miami, Florida and compares in situ measurements of the (size-resolved) concentration against aerosol reanalysis data. They bring to bear a wide variety of experimental and numerical tools, and the breadth of the analysis is impressive, making for overall robust results. The authors find that the surface concentration of dust at Miami is somewhat overestimated by the reanalysis products and is also finer than represented in these (highly similar) reanalysis products. This last conclusion is counter to findings by several other studies, including my own, so is surprising and possibly important. The article is well written and is overall excellent science. I do have some comments the authors should address before publication:

**Major comments:**
1. The conclusion regarding the overestimation of dust surface concentration by the reanalysis products is confusing in the abstract. Line 7 of the abstracts notes: "Measured near-surface dust mass concentrations slightly exceed model values" whereas line 13 notes: "model dust mass concentrations near the surface are higher than those measured". Please resolve these seemingly contradictory statements.

This is an oversight. The first statement is rewritten as "Most of the modeled dust mass resides in diameters between 2-6 micron, while most of the size-resolved measured dust mass is in diameters less than 2 micron." (also to avoid repetition).

2. A methodological problem is that the authors are comparing modeled dust size distributions in terms of geometric (volume-equivalent) diameter Dg against measured aerodynamic diameters Da. But because dust is both very aspherical (e.g., Okada et al., 2001; Kandler et al., 2007; Huang et al., 2020) and has a much larger density than water, a particle's geometric diameter is smaller than its aerodynamic diameter and a correction should be made. I recommend using the simple correction based on shape data in Kok et al. (2014), namely Dg = 0.75 Da.

[Figure]

Thank you very much for drawing our attention to this oversight! We have applied this simple correction, and have also replotted Fig. 9 as dM/dD - which we should have done for the initial submission of course. The new figure is shown to the left. The correction of the aerodynamically-measured diameters to a geometric

**Figure 9.** 17-day-average dust mass concentration per (one $\mu$m)$^{-1}$ diameter bin from measurements (blue) and GEOS-5 FP (black).

diameter increases the difference between the observed and modeled size-resolved dust mass concentrations. We have also rewritten section 4.4 to better describe the new figure.

3. I'm confused about GEOS-5 FP versus MERRA-2. The abstract describe these as "closely related" but section 2.4 ("GEOS-5 FP/MERRA-2") only includes a description of GEOS-5 FP and does not discuss MERRA-2. Please clarify the distinction (if any?) between these two products.

We mostly used the GEOS-5FP product for this project, as it was available prior to the MERRA-2 reanalysis. Both apply aerosol assimilation within the same model, but the GEOS-5FP aerosol product is at a 25km spatial grid spacing, while MERRA2 is global with a grid spacing of 50km. Some of the distinction is discussed on p. 2, lines 13-15 ("*The operational GEOS-5 Forward Processing (FP) model relies on a slightly more mature version of the GEOS-5 model used to produce the global Modern-Era Retrospective analysis for Research and Application-Version 2 (MERRA-2; Randles et al., 2017)*."

To better emphasize GEOS-5 FP/MERRA-2 distinctions within section 2.4, some language has been restructured into that section ("*The GEOS-5 FP/MERRA-2 products are similar enough that they are used interchangeably within this study, reflecting their incorporation at different times of the study*") and an additional sentence incorporated ("*Aerosol products from the operational GEOS-5 FP model are available at a 25 km grid spacing, slightly finer than the 50 km grid spacing of the global MERRA-2 reanalysis.*")

4. Data availability – the authors note data is available from the first author, but these data are valuable to the community and really ought to be posted on a publicly available repository.

This has been done. The data availability statement now includes: "*The datasets developed for this study (lidar extinction retrievals, size-resolved dust mass concentrations and measured sodium mass concentrations) are available from the University of Miami Scholarly Repository at https://doi.org/10.17604/1b5v-h184. These include the sun photometer aerosol optical depths, which are also publicly available through the AERONET website.*"

I think Figure 9 is the paper's most salient result – that surface dust in Miami is finer than represented in aerosol reanalysis products – but some corrections need to be made to the presentation. Panel c shows dust mass concentration versus diameter in terms of bars, and a reader would reasonably conclude that the surface area of the bar is representative of the mass of aerosol in that size class. But that is not the case here, and instead the total mass in each size class is plotted. This leads to a distortion of the data in that larger size bins will be correspondingly larger (e.g., if you divide the 4-10 um bin into two separate bins, then each bin would be only half the height of the current representation). To fix this, the authors need to normalize the mass concentration by the diameter range, which is equivalent to plotting dM/dD, as is standard (although they could also plot dM/dlnD and use a logarithmic x-axis).

Yes! We should have done this in the original submission of course. This has now been done.

**Minor comments:**
Line 27-29 on p. 4: "some reduction in aerosol asphericity may

also occur from chemical aging". Measurements indicate that chemical aging of North African dust is quite limited (Denjean et al., 2015), and Huang et al. (2020) just showed that North African dust becomes more aspherical during transport, probably because of preferential settling of spherical dust particles. So this statement should be adjusted.

*This has been rewritten as "A relatively consistent \delta_p throughout transport to the eastern Caribbean can reflect a lack of atmospheric processing (Denjean et al., 2015), in which the externally-mixed dust particles may even increase in asphericity (Huang et al., 2020), although less is known of the chemical composition and shape of dust arriving at Miami via the Gulf of Mexico and subsequent westward transport over southeastern United States (Kramer et al., 2020)."*

P. 11: "transport of at least 20,000 km". Earth's circumference is 40,000 km, so this seems incorrect.

Changed to '~6000 km'

Second column of Table 1 should specify that this is diameter.

Table 1 has also been modified to include an additional column indicating the geometric diameter corresponding to the impactor aerodynamic diameter thresholds.

Table 2. Length > width and aspect ratio >= 1, so the entries in the last column (which are all < 1) should be the reciprocal number. It'd be interesting to note how the measured aspect ratios compare to other literature data on dust shape, as compiled in Huang et al. (2020).

*The Table has been corrected, and a sentence added to page 9: "Although these are only five samples, their mean aspect ratio of 1.9 is equal to the median calculated from almost 78000 samples gathered in Puerto Rico (Reid et al., 2003a, b; Huang et al., 2020). This asphericity will contribute to the survival of the particles, as aspherical particles fall at slower terminal speeds than spheres of equivalent mass (Yang et al., 2013; Huang et al., 2020)."*

Figure 2. If there is no data, then no data should be plotted, so please remove the zeroes for the bulk mass concentration data (8/4-8/9). Done
Figure 3: please note units for dust mass concentrations in caption. Done.
Figure 7: please note whether correlation is taken in linear or logarithmic space (it ought to be the latter since the data spans several orders of magnitude). Done

**Reviewer 2**

The manuscript is clearly structured and in general terms well written. In few sections it lacks quantification of results, which requires modification. The manuscript can be accepted for publication in ACP after major revisions have been considered which are specified in the following. Revisions are classified major since the adjustment of diameters to either volume equivalent or aerodynamic equivalent may alter the main conclusions.

Thank you for the comments. Please note that the adjustment of the measured diameters to their geometric value increases the discrepancy between the modeled and measured diameters.

SPECIFIC COMMENTS

1. The abstract contains the conclusion "Measured near-surface dust mass concentrations slightly exceed model values, with most of the modelled dust mass in diameters between 2-6 µm." The data shown in Fig. 6 show the opposite behaviour. Please clarify. Done.

2. The results presented graphically in Figs. 5 -7 are described in the text only qualitatively as "robust correlation" (page 4, line 13), "slight underestimation" (page 7, line 22) or "clear correlation" (page 8, line 13). In all cases the results from a linear correlation analysis should be presented and discussed in the text.

Page 4, line 13, refers to Figure 2. The correlation coefficient between the 11 independent samples is 0.98. The text has been rewritten to include this.

Page 7, line 22 refers to Fig. 5. The correlation coefficient is 0.80 (for those comparisons for which AERONET values are available). This is now incorporated, in both the caption and the text.

Page 8, line 13 refers to Fig. 7, for which the correlation coefficient is 0.63. This value is now included within the text, where it is also discussed .

3. The presentation and discussion of dust particle sizes lacks clarity. Size distributions where measured by a cascade impactor which sorts particles according to their Stokes number and thus aerodynamic equivalent diameter which assumes unit density (1 g cm$_{-3}$). The models use spheroidal particles to describe dust particles with given densities. All values are listed in Table 1, but there is a conversion missing from aerodynamic equivalent diameter with density 1 g cm$_{-3}$ volume equivalent diameter with densities 2.5 to 2.65 g cm$_{-3}$. However, this conversion is essential for the comparison pf the results and needs to be included. Furthermore, the title of column 4 should be modified from "Miami" to, e.g., "Miami impactor cut-off diameter".

Thank you for pointing this out, as did Reviewer 1. We have expanded Table 1 to also include the geometric diameters estimated from the aerodynamic diameters and have redone Fig. 9, as shown on the first page of this response.

Figure 9 showing the results form the size distribution comparison need to be adjusted to the respective size classes after the diameters have been converted to one type (either volume equivalent or aerodynamic equivalent). Besides adjusting the diameters of one data set to the other, no matter which serves as reference, the size distributions need to be presented as dM/dlnD or dM/dlog D, or dM/dD, whatever is preferred. Otherwise the size distributions cannot be compared.

We have redone Fig. 9 (see page 1 of this response).

Finally, Fig. 12 lacks explanation. To my understanding, it shows the vertical profiles of mass mixing ratios for the 5 size classes of GEOS-5 FP, and thus the vertical distribution of the total dust mass mixing ratio in the model would be the sum of the five classes. If this is correct, what is then the "Average"? Here, some explanation is requested.

Thank you for catching this, the "average" line refers to a line from a different plot that is not referenced within the manuscript. We have removed this line.

MINOR ISSUES

Page 2, line 32: blank space between "a" and "MPL". Done.

Page 4, line 31: It might be easier to read if the equations are presented on a separate line. Done.

Page 6, line 25: The Chapter title "Overview" is misleading. The title should contain an indication that here the presentation of results starts, e.g., "Overview of dust seasons from 2014 to 2016".

Thank you for mentioning. We have retitled this section as "Overview of 2014-2016 summer dust seasons".

Section 4.4: Figure 10 seems to be introduced before Figure 9, please check. We were missing a reference to Fig. 9, now included.

Page 9, line 35: use lower –case letter after comma. Fixed.

Page 11, line 8: the term "$\boxed{?}_{aer}s$" is confusing, since it also may refer to the product of aerosol optical depth $\boxed{?}_{aer}$ and a property $s$. Please rephrase. I have removed the use of the plural throughout ( and simplified the subscript to just be an 'a').

Page 22, Figure 2: please remove the points from the plot where no data are available. The current plot is heavily misleading. Done.

Page 28, Figure 8: it would be of high relevance to the variability range of the average frequencies connected to the respective mass concentration bins. Please show -±1$\boxed{?}$ as error bars. These aren't average frequencies, they are the actual frequencies of the days with total dust mass concentrations falling within the specified thresholds. There's no standard deviation to plot here.

---

## Author Response (AR3)

Response to Editor Comment on "Apparent dust size discrepancy in aerosol reanalysis in north African dust after long-range transport"
Author(s): Samantha J. Kramer et al.
MS No.: acp-2020-1
MS Type: Research article
Iteration: Correction

**Editor Decision: Publish subject to technical corrections** (17 Jul 2020) by Andreas Petzold
Comments to the Author:
Dear Paquita

thank you very much for the careful revision of the manuscript.
It is my pleasure to accept the manuscript for publication in ACP.

The only technical correction which needs to be fixed, concerns the font size of the x-axis labels in Figure 2, please check.

Best wishes

Andreas

Response:

Thank you for the final feedback and acceptance of the manuscript. I have corrected the font size in Figure 2. In addition my co-authors Anne Barkley and Cassandra Gaston made some finalizing edits to Table 2 and Section 2.2 describing the electron microscopy. These are minor but increase the precision of the language.

Regards, Paquita Zuidema